# Serological Diagnosis of Flavivirus-Associated Human Infections

**DOI:** 10.3390/diagnostics10050302

**Published:** 2020-05-14

**Authors:** Didier Musso, Philippe Desprès

**Affiliations:** 1IRD, AP-HM, SSA, VITROME, IHU-Méditerranée infection, Aix Marseille Université, 13005 Marseille, France; 2Laboratoire Eurofins Labazur Guyane, 35 rue Lieutenant Brassé, 97300 Cayenne, French Guiana; 3INSERM U1187, CNRS UMR 9192, IRD UMR 249, Unité Mixte Processus Infectieux en Milieu Insulaire Tropical, Plateforme Technologique CYROI, Université de La Réunion, 97491 Sainte-Clotilde, La Réunion, France; philippe.despres@univ-reunion.fr

**Keywords:** antigenic cross-reactivity, arbovirus, diagnosis, *Flavivirus*, immunoassay, infection, serology

## Abstract

Arthropod-borne viruses (arboviruses) belonging to the *Flavivirus* genus of the *Flaviviridae* family, are a major public health threat in tropical and subtropical regions, and have recently become a medical concern in temperate zones. Most flaviviruses are classified as zoonotic viruses. Human flavivirus infections can be asymptomatic, responsible for unspecific symptoms in the first few days following infection, or responsible for severe complications potentially resulting in death. During the first days following symptom onset, laboratory diagnosis of acute human flavivirus infection is mainly based on molecular detection of the viral genome by RT-PCR methods, followed by the capture of specific antibodies using serological tests after the first week of infection. The detection of antibodies that have virus neutralizing activity can be used to confirm flavivirus infection. However, human flavivirus infections induce the production of cross-reactive antibodies, often making serology inconclusive. Indeed, serological diagnosis of flavivirus infection can be hampered by a patient’s history of flavivirus exposure, particularly in regions where multiple antigenically related flaviviruses co-circulate. We focus our mini review on conventional immunoassays that allow the diagnosis of major flavivirus-associated human infections in basic, routine and high-profile central health centers; and the interpretation of diagnostic serology tests for patients living within different epidemiological situations.

## 1. Introduction

More than five hundred arthropod-borne viruses (arboviruses) are registered in the International Catalog of Arboviruses (https://wwwn.cdc.gov/Arbocat/), 25% of them are confirmed human pathogens [1]. Most arboviruses of public health importance are members of the *Flaviviridae*, *Togaviridae* and *Bunyaviridae* families [2]. Within the *Flaviviridae* family, members of the *Flavivirus* genus have become an increased global health problem over the past decades due to the extension of their global distribution such as dengue virus (DENV) [3,4], West Nile virus (WNV) [5,6], or Zika virus (ZIKV) [7,8]. Reemergence in areas where circulation was previously thought to be contained has been observed, as for yellow fever virus (YFV) [4]. Other medically-important flaviviruses such as Japanese encephalitis virus (JEV), which circulates mainly in Southern and Southeastern Asia [9], or tick borne encephalitis virus (TBEV), which is endemic in parts of Eurasia [10], have not yet expanded their global distribution but they have the potential for spread because their vectors are widely distributed. To date we are still not able to predict emergence or re-emergence of flaviviruses in areas where competent arthropod vectors are established. Indeed, all areas with competent vectors should be prepared for the emergence or reemergence of flaviviruses and arboviruses in general [11]. Medically-important flaviviruses are an increasing concern in tropical and subtropical regions but also now in temperate zones. Travelers returning from areas of endemic flavivirus circulation and the transport of infected animals increase the likelihood of introducing a new pathogen into temperate regions where competent arthropod vectors are increasingly present [12]. This was the case with the emergence of WNV in North America [13] and Europe [14] from the 2000s; DENV [15] in Mediterranean countries for the last ten years or more recently ZIKV in South America in 2015 [16] and recently Europe in 2019 [17]. To prevent local transmission of newly introduced flaviviruses, early diagnosis of autochthonous infections and imported infections from travelers and their rapid notification have therefore become a priority in non-endemic areas. Newly emerging viral infections can be associated with the description of new modes of transmission, as for ZIKV that has been shown to be transmissible in humans through sexual intercourse [7,18]. Flavivirus-related human diseases resulting from non-vector-borne transmission, mainly blood transfusion for ZIKV [19], WNV [20] or DENVs [21] as well as sexual and mother-to-child transmission for ZIKV [7], can complicate the individual diagnosis in flavivirus endemic areas.

Flavivirus-associated human disease diagnosis requires laboratory testing, either through direct detection of the infecting agent, or the detection of antibodies directed against the infecting virus. The detection of viral RNA using PCR-based techniques offers excellent detection performance of flaviviruses in biological fluids. Most of the time, viral RNA amplification techniques still require clinical laboratories with advanced technology. Antibody tests detect antibodies directed against flavivirus antigens, mainly the envelope E protein, and, depending on the serological assay, can detect different immunoglobulin classes including IgM, IgG, and IgA. Interpretation of serological results can be challenging, principally due to an extensive cross-antigenic reactivity between the members of the *Flavivirus* genus [6,10,22,23,24,25,26,27,28,29]. In this review we discuss the place of serology in laboratory diagnosis of flavivirus-related human diseases, and the advantages and limitations of the main serological assays, with a focus on interpretation of serological results in different epidemiological settings. 

## 2. Laboratory Diagnosis of Flavivirus-Associated Human Diseases

Routine diagnosis of flavivirus-associated human diseases relies on the detection of the pathogen, its nucleic acids or specific viral antigens during the acute phase of the disease followed by the capture of specific antibodies at least one week after the infection. Advantages and limitations of each method and the window of detection are detailed in Table 1 and Figure 1.

## 3. Early Diagnosis of Flavivirus-Associated Human Diseases

Flavivirus-associated human diseases can be reliably confirmed by the detection of viral nucleic acids using conventional reverse transcription (RT)-PCR or real-time RT-qPCR. The molecular diagnosis of flavivirus infection is dependent on the presence of viral nucleic acids in the sampled biological material. The RT-PCR assay, which is used routinely on blood specimens, can also be used on urine samples for ZIKV [30,31] and WNV [32], and less frequently on other biological samples such as cerebrospinal fluids (CSF). The window for ZIKV detection differs between body fluid samples with virus being detectable in semen specimens for long period of time (up several months) [33]. Notably, the molecular tests required for flavivirus detection in blood donations differs from those that are effective in clinical testing due to sensitivity requirements.

Although RT-PCR is a sensitive and specific method that allows rapid and reliable diagnosis of acute phase flavivirus in human diseases, its efficacy is mainly limited to the acute phase of infection (a few days post infection), and can be impacted by briefness of the viremic period or a low viral load in the blood in patients infected with flaviviruses such as WNV [20] or ZIKV [34]. However, the rate of detection in blood can be increased by testing whole blood instead of serum from patients infected by flaviviruses, especially for WNV [35] and ZIKV [36]. Indeed, laboratory testing standards need to adapt for “suspected flavivirus”.

The early diagnosis of dengue can also be performed using immunocapture of viral protein NS1 in the bloodstream using commercially available kits [37]. Unfortunately, other NS1-based tests are not yet available for other flavivirus infections.

In addition, virus isolation methods in cell lines such as African green monkey Vero cells and *Aedes* mosquito C6/36 cells are well-recognized methods for the confirmation of flavivirus infections, but their use is usually restricted to reference laboratories [38].

## 4. Detection of Flaviviruses Antibodies in Clinical Specimens

### 4.1. First Line Serological Assays

Table 2 details those immunological assays that are used in routine practice and those that are commercially available for flavivirus serology. Various techniques can be used to detect specific antibodies directed against flaviviruses. For serum specimens, hemagglutination inhibition assay and complement fixation techniques have been progressively replaced by commercial immunochromatographic lateral flow strip tests, immunofluorescence assays and enzyme-linked immunosorbent assay (ELISA) (including Mac and Gag-ELISA for the capture of IgM and IgG, respectively) [39,40,41,42,43]. Development of “in-house” ELISA tests by high profile central health centers is required to verify unvalidated serological data or when no licensed kits for detection of a particular flavivirus are yet available. However, such methods are uncommonly standardized. Advantages and limitations of each method are also detailed in Table 2.

Specific immunoglobulin class M antibodies are usually detectable from the first week to 3 months post infection, and immunoglobulin class G antibodies are usually detectable from 2 weeks to several months or years. However, important variations can take place in the kinetics of antibody response depending on the biological assay used, viral antigens targeted for the antibody capture, immunological status in relation to a previous flavivirus exposure either by natural infection or vaccination, primary and secondary flavivirus-related human diseases, and individual immunological backgrounds.

Specific antibodies can be detected in CSF for flaviviruses responsible for severe neurological complications, especially tick-borne encephalitis virus (TBEV) [10], WNV [6,25] or JEV [26], and are usually detectable from the second week post infection when neurological disorders are diagnosed (from the first week for JEV [26]). The CSF-serum antibody index can be used to discriminate between blood-derived and brain-derived specific antibody fractions [10].

The main limitation of flavivirus serological assays is antibody cross-reactivity [6,10,22,23,24,25,26,27,28,29]. Unfortunately, the exact extent of cross-reactions between different flavivirus members is unknown, additionally it depends on the biological assay used for diagnosis and on the level of exposure to other co-circulating flaviviruses. Flavivirus-induced antibody cross-reactivity can also depend on the route of infection, as it has been reported for YFV with different cross reaction patterns among individuals previously exposed by vaccination or natural infection [27,44,45]. Amongst arboviruses, serological cross-reactions are not restricted to flaviviruses but are also reported for alphaviruses [46,47]. Cross-reaction between flaviviruses and alphaviruses, if any, should be very uncommon. However cross-reaction has been reported between flaviviruses and unrelated viruses, and between DENV and the novel betacoronavirus SARS-CoV-2 [48].

External quality assessment for WNV [49], DENV [50] or TBEV [51] diagnostics has demonstrated a large heterogeneity in laboratory performance, suggesting that all laboratories have not adopted rigorous control standards in order to provide comparable standardized flavivirus diagnosis results. 

For ZIKV serology, cross reactions using first line serological assays are so promiscuous that official guidelines recommend the confirmation of all positive and inconclusive results by confirmatory serological assays [24]. Indeed, IgM and IgG antibody detection methods commonly suffer from false-positive and false-negative detection rates, meaning that most serological results are presumptive but not confirmatory.

DENV-1 to DENV-4 infections display similar symptoms, and antibody cross-reactivity makes simple antigen-based discrimination difficult between these four serotypes [52]. Differentiation of DENV serotypes can be achieved at the acute phase of the disease using serotype specific RT-PCR [37]. Therefore, serological differentiation of infection with the different DENV serotypes is generally achieved by neutralization tests, as greater specificity is shown in antibody neutralization than in antibody recognition. Notably, individuals experiencing a secondary DENV infection are characterized by the presence of high titers of production of anti-DENV IgGs during the acute phase of infection due to an anamnestic response, whereas IgGs are usually detected 10–15 days after a primary DENV infection [52].

Serological assay alone cannot discriminate primary from secondary DENV infections known to cause more severe disease that could relate to antibody-dependent enhancement phenomena [52,53].

### 4.2. Confirmatory Serological Assays

The detection of neutralizing anti-flavivirus antibodies is correlated with the presence of the specific IgGs in blood specimens. Conventional plaque-reduction neutralization tests (PRNT) and virus neutralization assays (VNA) in microplates are considered the “gold standard” in discriminant flavivirus serology [54]. However, regional structural variation in flaviviruses means that PRNTs or VNAs should use local virus isolates, when possible, in order to achieve better specificity and sensitivity. This means that no standardized commercial materials can be used for the establishment of effective local diagnostic measures, and laboratory constraints mean that the large-scale application of such diagnostic measures is typically limited to centralized diagnostic services. Neutralization tests often necessitate manipulation of flaviviruses in a biosafety laboratory level (BSL) 2, 3 or 4 with respect to the biological risk classification, which differs among countries [55]. 

Although virus neutralization assays are the reference tests for serological diagnosis of flavivirus-associated human diseases, serological cross reactions between the flaviviruses are also reported with this assay [56] and variations in PRNT titers varied according to testing conditions [57]. Consequently, the development of new neutralizing assays is a priority for flavivirus-associated human disease diagnosis. Flow cytometry-based neutralization assays using GFP or Luc reporter gene virus are a promising method, taking a few days to complete and numerous clinical samples may be measured for neutralizing antibodies in the same run [58,59,60,61,62,63,64].

### 4.3. Innovative Serological Assays

One major advance in the diagnosis of flavivirus human-associated infections is the development of multiplex assays that can simultaneously detect antibodies directed against several flaviviruses in the same run. High-density microarrays containing peptides derived from various flaviviruses have been developed to improve the sensitive and specificity of flavivirus serodiagnosis [65]. Microsphere-based immunoassays (MIAs) for the measurement of IgM and IgG against a large variety of flaviviruses are now a technology of choice to provide a multiplex serological assay [66,67,68,69,70,71,72,73,74].

### 4.4. Serosurvey Studies

A wide range of methodological designs have been applied to serosurvey studies and the lack of standardization represents a major limitation in the interpretation of results, as demonstrated for DENV or ZIKV [75].

## 5. Challenges in the Serological Diagnosis of Human Flavivirus Infections

Diagnosis should take into account the access to laboratory tools, the epidemiology of arbovirus diseases in the region where the patient is supposed to have been infected, and the past exposure to flaviviruses.

### 5.1. Access to Laboratory Tools

Most countries that experience high levels of endemic flavivirus circulation have limited laboratory capabilities. It is not unusual that only rapid serological diagnosis test kits are available at the point of care testing [76,77]. Consequently, the number of licensed rapid test kits has been growing fast over recent years. For dengue disease, rapid tests offer a good specificity (usually up to 90%), but experience large limitations with a sensitivity ranging from 10 to 99% [78,79]. Combined immunocapture of soluble NS1 antigen with the capture of specific antibodies has been demonstrated to increase the effectiveness of rapid tests for dengue diagnosis [80].

In countries with limited laboratory capacities, ELISA assays are usually not available and there is no access to confirmatory virus neutralization assays. This is one of the main limitations of the implementation of international recommendations and guidelines that focus on gold standard assays but do not take test and facility availability into consideration.

In high income countries, even if tests are available and access to reference laboratory possible, the issue is that commercial serological assays have not been developed for each flavivirus of medical concern. 

### 5.2. Co-Circulation of Flaviviruses 

Multiple flaviviruses co-circulate in most areas of endemic diseases [81], and co-circulation patterns vary over time and space [82]. Indeed, local epidemiology and the timings of potential exposure should always be taken into consideration when interpreting results. 

Another consequence of co-circulation is the high risk of co-infection and a positive result for a flavivirus does not preclude infection with another one [83].

In endemic areas of several flaviviruses, the rate of serologic positivity against such pathogens is so elevated that most serological assays are unreliable for the diagnosis of a flavivirus infection. This becomes even more intricate in regions where confirmatory serologic diagnosis tests are still lacking or when most people have been vaccinated for flavivirus-associated diseases such as dengue, yellow fever and Japanese encephalitis. Implementation of multiplex microsphere immunoassays (MIA) using native or recombinant viral antigens could be appropriate for resolving the complexity of flavivirus diagnostic serology in endemic areas [74,84,85,86].

### 5.3. Prevalence of Flaviviruses

Most serological assays against flaviviruses are evaluated in terms of specificity and sensitivity, but not in predictive value, which is the probability that the test gives the correct diagnosis [87,88,89]. Positive and negative predictive values (PPV and NPV) are subject to variations according to the prevalence of the disease in a population. If a test is 99% specific, and 10% of a population being tested have the disease, there will be only 1 false positive for 10 true positives. However, if only 1 out of 1000 people experiences the disease, there will be 10 false positives for 1 true positive. In other words; as the prevalence increases (e.g., epidemic situation), the PPV also increases but the NPV decreases, similarly, as the prevalence decreases (e.g., low endemic situation), the PPV decreases while the NPV increases. The evaluation of the test should use sample cohorts relevant for the local setting and only the test evaluated in this setting should be used. 

As mentioned above, because the prevalence of flavivirus-associated human diseases is highly variable in time and space in endemic areas, interpretation of results requires detailed knowledge of the local epidemiology at the moment the patient has been exposed.

### 5.4. Determination of the Onset of Symptoms

As the window of detection of flavivirus RNA, of specific IgM and IgG antibodies, is determined relative to the onset of symptoms, determination of the exact timing of the symptoms is key to using the correct diagnostic test. Usually the onset of symptoms can be clearly identified in infections such as dengue because there is a rapid onset of high fever. However, in some flavivirus infections such as Zika, there is often no fever or a low grade fever without acute onset of symptoms [8]. If there are any doubts, both nucleic acid testing and serology should be performed.

### 5.5. Patient’s Past Exposure to Flaviviruses

As most flavivirus-associated human diseases are asymptomatic or responsible for mild symptoms, patients may have past unrecognized disease. Indeed, in endemic areas it should be considered that all residents may have cross-reactive antibodies.

In flavivirus endemic regions where the populations are engaged in a vaccination program such as yellow fever in Africa and South America and Japanese encephalitis in South East Asia, or enrolled in a clinical trial of a candidate vaccine against a flavivirus of medical interest, interpretation of serological testing necessitates available information on the patient’s vaccination history.

### 5.6. Interpretation of Serological Results in Endemic Areas and in Travelers Returning from Endemic Areas

To illustrate the complexity in interpreting flavivirus serological results, we will consider the example of French Polynesia (South Pacific region), which suffers from high levels of flavivirus circulation, is a major tourist area, and has been identified as a hub for the spread of medically-important flaviviruses including Zika in 2013–2014 [70].

DENV was the only flavivirus circulating in French Polynesia until the emergence of ZIKV in 2013. DENV and ZIKV co-circulated from 2013 to 2014, and since mid-2014 DENV is the only circulating flavivirus [90].

Until the emergence of ZIKV in French Polynesia, the available serology tests for DENV provided reliable diagnosis for the disease among a large majority of residents. During the period of co-circulation of DENV and ZIKV, there was great difficulty in the serological diagnosis of both diseases due to a strong cross-antigenic reactivity between the two flaviviruses.

At the end of the ZIKV epidemic in French Polynesia, where at least one-half of the residents developed ZIKV antibodies [67], the serological diagnosis of DENV infection remained unreliable due to residual cross-reactive antibodies in individuals previously exposed to either flaviviruses [67].

For a traveler returning from French Polynesia, the situation is quite different. As has been observed with residents, a serology positive for DENV was specific prior to the emergence of ZIKV, whereas after the emergence, interpretation was confounded for the whole period of virus co-circulation due to the risk of cross-reactive antibodies. However, after the period of co-circulation, a positive serology could once again be considered specific for DENV (excepted if the traveler has a history of travel to another endemic area for flaviviruses or has been vaccinated against another flavivirus).

Indeed, the same serological test can have high or low predictive values, depending on the circulating rate of flaviviruses in the concerned area at the time of infection.

## 6. Conclusions

Flavivirus-associated human diseases are a growing cause of morbidity and mortality across the world and their area of distribution is expanding. Flaviviruses show a large amount of cross-reactivity in serological diagnostic tests, triggered by natural infection or vaccination. Cross-antigenic reactivity combined with a large overlap in clinical syndromes, co-circulation of different flaviviruses, and poor access to advanced laboratory diagnosis tools for serological confirmation make serological diagnosis of flavivirus-associated human diseases a great challenge. Most international guidelines and recommendations rely on gold standard assays, such as the virus neutralization assay, that unfortunately are not available in most areas with high endemic flavivirus circulation. We have identified an urgent need for more specific and sensitive serological methods allowing reliable standardized diagnosis for the timely management of flavivirus-associated human diseases.

Due to the frequent unreliability of current flavivirus serological tests, it is strongly encouraged to use molecular diagnosis in the first days following the onset of symptoms. However, we are convinced that implementation of multiplex serological assays based on the use of recombinant viral antigens will open avenues for the development of new specific and sensitive diagnostic tools for flavivirus detection in the near future [66,67,68,69,70,71,72,73,74].

## Figures and Tables

**Figure 1 diagnostics-10-00302-f001:**
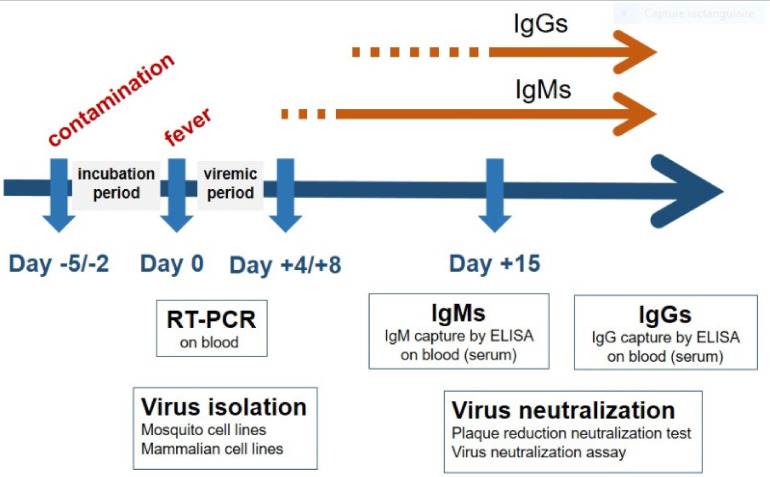
Flow chart of molecular and serological diagnosis tests in the course of human flavivirus infections.

**Table 1 diagnostics-10-00302-t001:** Current laboratory techniques for the diagnosis of acute human flavivirus infections.

Methods	Advantages	Critical Evaluation
RT-PCR	Diagnosis is performed by detection of viral nucleic acids. Specificity and sensitivity. Rapidity. RT-PCR diagnosis test kits.	Positivity often limited to the acute stage of disease (2–7 days). Flavivirus infection can cause a weak or no viremia.
Virus isolation	Direct pathogen detection plays a key role in diagnosing flavivirus infection	Biosafety Laboratory considerations (BSL levels 2 to 4). Requirement of cultured cell lines for viral growth. Virus identification using specific detection tools. Time consuming.
Viral antigen capture	Diagnosis of acute dengue virus infection based on soluble NS1 capture. Rapid diagnosis test kits.	Only available for dengue. False-dengue positivity has been documented.
Serology	Diagnosis is performed by IgM and IgG capture or virus neutralization assays. Qualitative and quantitative serologic diagnosis tests. Licensed rapid serologic diagnosis test kits.	Specificity and sensitivity. Complexity of serological flavivirus diagnosis. False interpretation of dengue diagnostic serology tests during secondary dengue infection. Virus neutralization assays require BSL levels 2 to 4. Serological assays performed in BSL are time consuming. Detection of antibodies does not exclude convalescent patients who may develop other illnesses with similar symptoms

**Table 2 diagnostics-10-00302-t002:** Serological methods for detecting flavivirus-specific antibodies.

Serological Diagnostic Methods	Virus-Specific Antibody Detection	Licensed Test Systems	Automation	General Remarks on the Method
Immuno-fluorescence tests (IFT)	Yes	Yes	Available IFT automation	IFT are referred as a conventional serological method
Enzyme-linked immunosorbent assays (ELISA)	Yes	Yes	Available ELISA automation	Conventional ELISA assays detect and measure a single analyte per plate
Virus neutralization tests (VNT)	Yes	No	No	VNT are performed to confirm the results of conventional serological methods
Lateral-flow immunoassays	Yes	Yes (only for DENV)	No	Rapid diagnostic tests but sensitivity and specificity are usually lower than for other methods.
Multiplex immunoassay (MIA)	Yes	No	Yes (MIA is based on the flow-cytometry technology)	MIA permits the multiplexing of many different assays within a single sample

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
