# Peer review of "Serological Diagnosis of Flavivirus-Associated Human Infections"

_diagnostics, 2020, doi:10.3390/diagnostics10050302_

Round 1
Reviewer 1 Report
This is an important topic to review at the present time and the authors appear have a lot of experience in the diagnosis of flaviviral diseases. More coherent presentation with a clear focus and better written English will make the article more readable and useful to the field.
- Lab resources are quite variable and often there are well resourced private labs in poorer countries. It is archaic and inaccurate now to classify countries into LMIC, HIC for purposes of available lab facilities.
- Suggest reorganising article into describing major serological techniques in overview, and how they are used in major flaviviral diseases, and an outline of the facilities required.
- qPCR is being undervalued and authors might be more circumspect about the sensitivity of detection - details on PPV/NPV and disease prevalence is basic and may be omitted.
- The mss should discuss the option that cross-recognition of antigens is so prevalent that serodiagnosis of any one flaviviral disease, unless there is an epidemic of that disease, is impossible where many flaviruses are circulating unless specific peptide epitopes can be identified [seems very difficult even with sophisticated multiplexing]. What is the way forward here?
- How common is cross-immunity in neutralisation assays?
- Can Fig1 and Table1 be combined into one better illustration?
- Table 2 needs to be improved.
- Is there cross-reactivity with other groups of arboviruses ?
- Reactivity with DENV NS1 rapid tests has now been seen in SARS2.
Author Response
Reviewer 1:
This is an important topic to review at the present time and the authors appear have a lot of experience in the diagnosis of flaviviral diseases. More coherent presentation with a clear focus and better written English will make the article more readable and useful to the field.
The entire text has been read by a native English speaking colleague (New–Zealand) whom has all expertise in the field of this work.
- Lab resources are quite variable and often there are well resourced private labs in poorer countries. It is archaic and inaccurate now to classify countries into LMIC, HIC for purposes of available lab facilities.
We have reworded for “countries with limited laboratory capacities”.
- Suggest reorganising article into describing major serological techniques in overview, and how they are used in major flaviviral diseases, and an outline of the facilities required.
We appreciate this point raised by the reviewer #1. In the revised manuscript, we have modified the Table 1 which now better describes the different laboratory techniques of the diagnosis of human flavivirus infections with a focus on the biosafety laboratory facilities required. Consequently, Table 1 and Figure 1 have been kept in the state. We also improved table 2, lastly, the plan of the manuscript has not been changed. The entire text has been read by a native English speaking colleague whom has all expertise in the field of this work.
- qPCR is being undervalued and authors might be more circumspect about the sensitivity of detection - details on PPV/NPV and disease prevalence is basic and may be omitted.
We agree that details on PPV/NPV are basic because it is not the topic of our manuscript. However, from our experience, a lot of physicians ignore these concept. Unfortunately they consider that is a serological or a PCR assay is positive it means that the patient has the disease, and that if the results if negative it means that the patient has not. We prefer to keep this part of the manuscript because it reflect our experience in the field.
- The mss should discuss the option that cross-recognition of antigens is so prevalent that serodiagnosis of any one flaviviral disease, unless there is an epidemic of that disease, is impossible where many flaviruses are circulating unless specific peptide epitopes can be identified [seems very difficult even with sophisticated multiplexing]. What is the way forward here?
We greatly thank reviewer #1 for his/her pertinent remarks. Consequently, the following sentences have been added in the main text: « In endemic areas for several flaviviruses, the rate of serologic positivity against such pathogens is so elevated that most of serological assays are unreliable for the diagnosis of a flavivirus infection. This becomes even more intricate in regions where confirmatory serologic diagnosis tests are still lacking or when most of people have been vaccinated for flavivirus-associated diseases such as dengue, yellow fever and Japanese encephalitis. Implementation of multiplex microsphere immunoassays (MIA) using native or recombinant viral antigens could be appropriated for resolving the complexity of flavivirus diagnostic serology in endemic areas (Tyson et al. PloS Negl.Trop.Med. 2019, doi.org/10.1371/journal.pntd.0007649; Taylor et al. Viruses, 10:255, 2018; Wong et al. EBioMedicine 16:136, 2017; Cao-Lormeau et al. Lancet 387:1531, 2016)
- Can Fig1 and Table1 be combined into one better illustration?
We have modified Table 1 modified according to the reviewer’s comments #2. Consequently, Table 1 and Figure 1 have been kept as individualized ones in the revised manuscript.
- Table 2 needs to be improved.
We appreciate this point raised by the reviewer #1. In the revised manuscript, we have modified the Table 2 which now better describes the different serological diagnosis methods.
- IIs there cross-reactivity with other groups of arboviruses ?
We have included the sentences” “Amongst arboviruses, serological cross-reactions are not restricted to flaviviruses but also reported for alphaviruses 46,47 . Cross-reaction between flaviviruses and alphaviruses, if any, should be very uncommon. However cross-reaction have been reported between flaviviruses and unrelated viruses as between DENV and the novel and the novel betacoronavirus SARS-CoV-2”. And included 2 reference related to cross-reaction within alphaviruses and another one related to cross reactions between DENV and SARS-Cov-2..
- Reactivity with DENV NS1 rapid tests has now been seen in SARS2. See comment above for cross-reactions in serology
Reviewer 2 Report
The authors have described thoroughly the testing systems for flaviviruses. The paper needs quite a bit of editing to correct phrases and grammar. Here are the main examples:
14 Responsible to -> responsible for
17 of the viral genome
24 major flavivirus-related human diseases [Please correct throughout the manuscript]
37 areas where they were...
40 have not yet expanded
43 prepared for the emergence
44 arboviruses in general
46 flavivirus-endemic...
53 Newly emerging viruses...
57 living in
58 endemic areas
60 Detection... Please improve this sentence
63 flavivirus antigens [Please correct throughout the manuscript]
64 depending on [Please correct throughout the manuscript]
Table 1 3 or 4 pathogens
Figure 1 Flow chart
97 African green monkey...
104 flavivirus antibodies [Please correct throughout the manuscript]
Table 2 Please fit the column titles into the space
Table 2 depends from -> depends on [Please correct throughout the manuscript]
115 Kinetics...
120 TBE -> TBEV
125 in addition
139 disease, making...
144 high rate of anti... -> high rate of production of anti...
149 Delete “It is usual to note that”
161 as a result of varied...
167 A major advance in the diagnosis of flaviviruses...
176 This line should not be all caps
187 when using a combined test...
196 Flaviviruses co-circulation -> Co-circulation of flaviviruses
200 Please reduce the font
201 Prevalence of flaviviruses
204 values, which...
211 and only the test evaluated...
227 In flavivirus endemic regions
232 in flavivirus serological results
243 due to remaining cross-reactive...
249 Please clarify the sentence
242 the same serological...
Ref 3 (1997)
The article titles in the references should be lower case (refs 16, 26, 39, 42, 46, 51, 53, 55, 56, 61, 62, 74, 76, 81, 84)
Ref 37 Health
Author Response
Reviewer 2:
The authors have described thoroughly the testing systems for flaviviruses. The paper needs quite a bit of editing to correct phrases and grammar. Here are the main examples:
14 Responsible to -> responsible for: Corrected
17 of the viral genome: Corrected
24 major flavivirus-related human diseases [Please correct throughout the manuscript]: Corrected
37 areas where they were...: Corrected
40 have not yet expanded: Corrected
43 prepared for the emergence: Corrected
44 arboviruses in general: Corrected
46 flavivirus-endemic...: Corrected
53 Newly emerging viruses...: Corrected
57 living in: Corrected
58 endemic areas: Corrected
60 Detection... Please improve this sentence Corrected
63 flavivirus antigens [Please correct throughout the manuscript]: Corrected
64 depending on [Please correct throughout the manuscript]: Corrected
Table 1 3 or 4 pathogens: arboviruses can be classified in levels 2, 3 or 4
Figure 1 Flow chart: Corrected
97 African green monkey...: Corrected
104 flavivirus antibodies [Please correct throughout the manuscript]: Corrected
Table 2 Please fit the column titles into the space: Corrected and orientation of the table changed because of lack of space
Table 2 depends from -> depends on [Please correct throughout the manuscript]: Corrected
115 Kinetics...: Corrected
120 TBE -> TBEV: Corrected
125 in addition: Corrected
139 disease, making...: Sentence reworded
144 high rate of anti... -> high rate of production of anti...: Corrected
149 Delete “It is usual to note that”: Deleted
161 as a result of varied...: Corrected
167 A major advance in the diagnosis of flaviviruses...: Corrected
176 This line should not be all caps: Corrected
187 when using a combined test...: Corrected
196 Flaviviruses co-circulation -> Co-circulation of flaviviruses: Corrected
200 Please reduce the font: Corrected
201 Prevalence of flaviviruses: Corrected
204 values, which...: Corrected
211 and only the test evaluated... Corrected
227 In flavivirus endemic regions Corrected
232 in flavivirus serological results Corrected
243 due to remaining cross-reactive... Corrected
249 Please clarify the sentence Reworded
252 the same serological... Corrected
Ref 3 (1997)
The article titles in the references should be lower case (refs 16, 26, 39, 42, 46, 51, 53, 55, 56, 61, 62, 74, 76, 81, 84) Corrected
Ref 37 Health
Round 2
Reviewer 1 Report
This manuscript has been considerably improved and almost ready for publication
- A few queries in the attached mss need to be addressed in a revision.
- Some suggestions have been made to further improve English and a few scientific points that need addressing are also highlighted.

Author Response
Dear Editor and reviewers,
We thank the reviewer for fast review of our manuscript and for the suggested corrections and improvements.
As requested, we submit a revised manuscript, including all corrections.
We feel that the changes have improved the manuscript significantly and we hope you will accept this version for publication.
On behalf of all the authors,
Sincerely,
Dr Didier Musso, corresponding author